# Molecular Insights into Intramuscular Unsaturated Fatty Acid Deposition in Lambs Through Multi-Omics Profiling

**DOI:** 10.3390/ani15172617

**Published:** 2025-09-06

**Authors:** Xuewen Han, Xudong Liu, Yu Fu, Jinlin Chen, Cuiyu Lai, Xiaofan Yang, Xuesong Shan, Yang Chen, Huaizhi Jiang

**Affiliations:** College of Animal Science and Technology, Jilin Agricultural University, Changchun 130118, China; hanxuewen1123@163.com (X.H.); 19845486668@163.com (X.L.); 15164460626@163.com (Y.F.); chenjinlin7314@163.com (J.C.); 19843927673@163.com (C.L.); jianghz6806@126.com (H.J.)

**Keywords:** unsaturated fatty acids, lamb, transcriptome, metabolome, muscle lipid metabolism, muscle fiber type

## Abstract

**Simple Summary:**

Fat in meat plays an important role in its flavor and nutritional value. Among different types of fat, unsaturated fatty acids are considered healthier for humans. Improving the content of these healthy fats in lamb could benefit both consumers and the livestock industry. In this study, we selected two groups of lambs with high and low levels of unsaturated fatty acids in their muscles. We examined their gene activity, chemical compounds in the muscle, and the types of muscle fibers they had. We found that lambs with more unsaturated fat had different patterns of gene activity related to fat metabolism, energy production in cells, and immune function. We also detected changes in chemical compounds linked to fat and amino acid processing. A key finding was that the *MYH7* gene, which is related to the development of slow muscle fibers, was more active in lambs with higher unsaturated fat. These lambs also had more slow-twitch muscle fibers. Our findings provide new insights that could help farmers produce lamb with healthier fat, improving the quality of meat for consumers.

**Abstract:**

Unsaturated fatty acids are key contributors to the nutritional and sensory quality of lamb meat. To investigate the molecular basis of intramuscular unsaturated fatty acid variation, we selected lambs with divergent fatty acid profiles and performed integrated transcriptomic and untargeted metabolomic analyses of the longissimus dorsi muscle. The high unsaturated fatty acid group exhibited distinct gene expression patterns in pathways related to lipid metabolism, mitochondrial function, and immune responses. Metabolomic profiling revealed significant enrichment of metabolites involved in both the biosynthesis and degradation of fatty acids. Among the differentially expressed genes, *MYH7* was markedly upregulated in lambs with higher unsaturated fatty acid content, suggesting a potential regulatory role in energy metabolism or lipid homeostasis. These findings provide new molecular insights into the mechanisms underlying unsaturated fatty acid deposition in lamb and identify *MYH7* and other candidates as potential targets for improving meat quality through breeding or nutritional strategies.

## 1. Introduction

Mutton, a traditional dietary staple in many cultures, is prized for its distinctive flavor and rich nutritional profile, making it an important component of the human diet. In addition to its palatability, mutton is an excellent source of high-quality protein, vitamins, and trace elements, while containing relatively low levels of fat and cholesterol, thus offering significant health benefits compared with pork [1]. Mutton provides 25 to 30 g of complete protein per 100 g, which is essential for tissue growth, repair, and immune function [2,3]. Furthermore, mutton is abundant in B vitamins, including B12, B6, riboflavin, and niacin, as well as important minerals such as iron, zinc, and phosphorus. These nutrients play a crucial role in maintaining overall health and preventing anemia [4,5,6]. Consequently, global demand for mutton has grown rapidly in recent years [7].

Evaluations of mutton quality usually include physicochemical parameters such as pH, water-holding capacity, marbling, tenderness, cooking loss, and drip loss, together with nutritional attributes [8,9]. Among these parameters, the composition of intramuscular fat—especially the content and profile of unsaturated fatty acids (UFAs)—has garnered increasing attention. Mutton fat is primarily composed of monounsaturated fatty acids (MUFAs) and polyunsaturated fatty acids (PUFAs) [10]. These fatty acids not only contribute to the flavor of the meat but also offer positive health benefits. Diets high in UFAs are linked to lower levels of serum low-density lipoprotein (LDL) cholesterol and reduced cardiovascular risk. In particular, MUFAs such as oleic acid have been shown to improve lipid profiles and decrease atherosclerosis risk [11,12,13]. Thus, UFAs are critical determinants of both the nutritional value and sensory quality of meat, and in mutton specifically, their abundance and composition directly influence health outcomes and consumer ac-acceptance [14,15].

In response to increasing consumer demand for healthier meat products, enhancing UFA deposition in muscle has become a major objective in both sheep breeding and nutritional strategies. However, the regulatory mechanisms underlying UFA accumulation in skeletal muscle remain poorly defined and appear to involve complex networks of lipid synthesis, transport, oxidation, and storage [16,17]. Some well-known lipid metabolism enzymes, such as FASN, SCD, and ELOVLs, are considered important regulatory factors [18,19], and our previous research suggests that muscle fiber composition may also play a significant role [20]. Myosin heavy chain 7 (MYH7), a structural gene encoding the β-myosin heavy chain specific to slow-twitch fibers, is essential for determining fiber type and regulating metabolic characteristics in muscle tissue [21]. While MYH7 is classically associated with contractile function, its role in lipid metabolism has not been extensively investigated.

Currently, multi-omics approaches integrating transcriptomic, proteomic, and metabolomic data have provided powerful tools for dissecting the molecular basis of meat quality traits in livestock. In cattle and pigs, such strategies have revealed key regulatory genes and metabolites associated with fat deposition, muscle development, and flavor compound formation [22,23]. For instance, integrated transcriptomic and metabolomic profiling has linked fatty acid metabolism to flavor differences in beef and identified lipid–amino acid interactions critical for pork quality [24,25].

By contrast, studies applying multi-omics profiling to sheep remain relatively limited. Existing reports mainly focus on growth performance, wool traits, or general meat quality parameters, with fewer efforts directed at elucidating the molecular mechanisms underlying intramuscular fatty acid deposition [26]. Although some transcriptomic analyses have identified candidate genes related to lipid metabolism in sheep muscle, they often lack validation through metabolomics or integration with biochemical data [27]. Consequently, a comprehensive understanding of the regulatory networks that drive UFA accumulation in lamb meat is still missing.

In this study, we aimed to fill this gap by conducting an integrative transcriptomic and metabolomic analysis of longissimus dorsi tissues from lambs with divergent UFA contents. By combining differential expression, functional enrichment, and metabolic profiling, we particularly focused on the potential regulatory role of *MYH7*, given its established link to muscle metabolism. Our findings provide novel insights into the molecular basis of UFA deposition in lambs and highlight *MYH7* as a potential genetic marker for improving mutton quality.

## 2. Materials and Methods

### 2.1. Animals and Sample Collection

A total of 60 six-month-old crossbred female lambs (Small-tailed Han × Australian White), reared under standardized management conditions at Guofeng Livestock Farm in Changling County, were randomly selected for this study. The animals were housed indoors with ad libitum access to a nutritionally balanced diet, which included Leymus chinensis (50.0%), corn (20.0%), soybean meal (5.4%), DDGS (7.0%), soybean oil (15.0%), CaHPO_4_ (0.5%), limestone (1.4%), NaCl (0.5%), and vitamin and mineral additives (0.2%). Nutrient levels were formulated to meet the requirements for growing lambs, with metabolic energy at 11.99 MJ/kg, crude protein at 10.1%, calcium at 0.65%, and total phosphorus at 0.33%. Pre-slaughter body weights ranged from 30 to 50 kg, showing a normal distribution without outliers. All lambs were fasted for 24 h with ad libitum access to water before slaughter. Following exsanguination via the jugular vein, samples of the longissimus dorsi muscle were immediately collected. Samples for meat quality assessment were stored at 4 °C, while those for transcriptomic, metabolomic, and quantitative PCR analyses were snap-frozen in liquid nitrogen and stored at −80 °C. Samples intended for immunohistochemistry and immunofluorescence were fixed in 4% paraformaldehyde.

### 2.2. Meat Quality Traits and Fatty Acid Composition

Meat quality parameters were assessed as previously described by Fu et al. [20]. Briefly, longissimus dorsi muscle samples were collected to measure intramuscular fat content (IMF), pH, meat color (L*, a*, b*), shear force (SF), and cooking loss. IMF was determined using the Soxhlet extraction method. Ph 24 was measured within 2 h postmortem at the thoracolumbar junction after homogenization in 0.1 mol/L KCl. Meat color was evaluated after 20 min of blooming using a CR-400 chromameter (Konica Minolta, Tokyo, Japan) with an 8 mm aperture under D65 illumination. Shear force was quantified using a TA-XT Plus texture analyzer (Stable Micro Systems, Godalming, UK) on cooked cylindrical samples aligned parallel to muscle fibers. Cooking loss was calculated by weighing standardized meat cubes (2 × 2 × 2 cm, ~30 g) before and after heating to 70 °C in a water bath. All measurements were performed in triplicate and averaged. Fatty acid composition was analyzed by liquid chromatography following the Chinese National Food Safety Standard GB 5009.168-2016 [28].

### 2.3. Transcriptome Sequencing and Analysis

Total RNA was extracted from longissimus dorsi muscle tissues of 6 lambs (n = 6 per group, high UFA vs. low UFA) using TRIzol reagent (Invitrogen, Carlsbad, CA, USA), following the manufacturer’s protocol. RNA purity and concentration were assessed using a NanoDrop 2000 (Thermo Scientific, Waltham, MA, USA), and integrity was evaluated using an Agilent 2100 Bioanalyzer (RIN > 7.0). mRNA sequencing (mRNA-seq) libraries were prepared using the NEBNext^®®^ Ultra™ RNA Library Prep Kit (NEB, Ipswich, MA, USA) and sequenced on the Illumina NovaSeq 6000 platform with 150 bp paired-end reads. Clean reads were aligned to the sheep reference genome Oar_rambouillet_v1.0 (GenBank assembly accession: GCA_002742125.1) [29] using HISAT2 v2.2.1 (Johns Hopkins University, Baltimore, MD, USA) [30], and gene expression levels were quantified as fragments per kilobase of transcript per million mapped reads (FPKM). Differentially expressed genes (DEGs) were identified using DESeq2 v1.30.1 (Bioconductor, Fred Hutchinson Cancer Research Center, Seattle, WA, USA) [31], with FoldChange > 1.5 and *p* < 0.05 as thresholds. Functional enrichment analyses including Gene Ontology (GO) and KEGG pathway enrichment were conducted using the ClusterProfiler package v4.0.5 (Bioconductor, Yu Lab, Southern Medical University, Guangzhou, China) [32] in R (R Foundation for Statistical Computing, Vienna, Austria), with Ovis aries Oar_rambouillet_v1.0 as the reference genome background.

### 2.4. Metabolomics Analysis

Untargeted metabolomics profiling was conducted on longissimus dorsi tissues (n = 6 per group) using ultra-performance liquid chromatography coupled to tandem mass spectrometry (UPLC–MS/MS). Tissues were homogenized in 80% methanol, followed by centrifugation and filtration. Chromatographic separation was performed using a Waters ACQUITY UPLC system equipped with a BEH Amide column, and metabolites were detected using a QTRAP 6500+ mass spectrometer (AB Sciex). Quality control samples were included throughout the run. Peak extraction, alignment, and quantification were conducted using Analyst software v1.7.1. Metabolites were identified and annotated based on *m*/*z* values and MS/MS fragmentation patterns using the HMDB databases. Metabolites with variable importance in projection (VIP) > 1 and *p* < 0.05 (*t*-test) were considered significantly different. Multivariate statistical analyses, including principal component analysis (PCA), were conducted using MetaboAnalyst v6.0 (Xia Lab, McGill University, Montreal, QC, Canada) [33].

### 2.5. Quantitative Real-Time PCR (qPCR)

Total RNA (1 µg) was reverse-transcribed into cDNA using the PrimeScript RT Reagent Kit (Takara, Tokyo, Japan). Quantitative PCR was performed with TB Green Premix Ex Taq II (Takara) on a CFX96 Real-Time PCR System (Bio-Rad, Hercules, CA, USA). Primer sequences were designed using Primer-BLAST (NCBI) and are listed in Appendix A. PCR conditions were as follows: initial denaturation at 95 °C for 30 s, followed by 40 cycles of 95 °C for 5 s and 60 °C for 30 s, with a melt-curve analysis performed at the end of amplification. Gene expression was normalized to β-actin, and relative quantification was calculated using the 2^−ΔΔCt^ method. Each reaction was run in three technical replicates and six biological replicates.

### 2.6. Western Blot Analysis (WB)

Total protein was extracted from muscle tissues using RIPA lysis buffer with protease inhibitors (Beyotime, Shanghai, China). Protein concentrations were measured using a BCA Protein Assay Kit (Thermo Scientific). Equal amounts of protein (20 μg) were separated by 10% SDS-PAGE and transferred to PVDF membranes (Millipore, Billerica, MA, USA). After blocking with 5% non-fat milk in TBST, membranes were incubated overnight at 4 °C with primary antibodies against FADD, MYH7, TMEM74, CSNK2A2, CTSB, and β-actin (loading control), followed by HRP-conjugated secondary antibodies. Immunoreactive bands were visualized using enhanced chemiluminescence (ECL, Thermo), and band intensities were quantified using ImageJ v1.54g. Each group included three biological replicates. Antibody details are listed in Appendix A.

### 2.7. Gene Set Enrichment Analysis (GSEA)

During the differential gene analysis, *MYH7* attracted our attention. To highlight the mechanism of *MYH7*-mediated UFA deposition, we performed a single-gene GSEA using *MYH7* as the identifier. Gene expression matrices were input into GSEA software v4.1.0 (Broad Institute, Cambridge, MA, USA), and enrichment was assessed using KEGG and Hallmark gene sets from MSigDB. Samples were divided into high- and low-expression groups based on the median expression of the target gene. Normalized enrichment score (NES), nominal *p*-value, and false discovery rate (FDR q-value) were calculated, with significance defined as FDR < 0.25 and *p* < 0.05.

### 2.8. Statistical Analysis

Data are presented as mean ± SEM. Differences between high and low slow-twitch groups were analyzed using unpaired *t*-tests with SPSS Statistics v27.0 (IBM Corp., Armonk, NY, USA). Statistical significance was set at *p* < 0.05, and highly significant differences at *p* < 0.01. Graphs were generated using GraphPad Prism 9.5 (GraphPad Software, San Diego, CA, USA).

## 3. Results

### 3.1. Comparison of Meat Quality Traits and Fatty Acid Composition Between High- and Low-UFA Groups

To investigate the potential mechanisms underlying intramuscular unsaturated fatty acid (UFA) variation, six lambs with the highest UFA content (H) and six with the lowest (L) were selected from a cohort of sixty individuals based on fatty acid profiles of the longissimus dorsi muscle. As shown in Figure 1, the H group exhibited significantly higher UFA levels than the L group (*p* = 0.0001), while body weight showed no significant difference between groups (*p* = 0.8915), suggesting that UFA variation was independent of growth traits and that the selected samples were suitable for subsequent comparative analyses.

Meat quality traits were further assessed (Figure 1). Among these, the H group showed a significantly lower intramuscular fat content compared to the L group (*p* = 0.0117) and a significant increase in muscle moisture (*p* = 0.0459). Other parameters, including pH24, shear force, cooking loss, meat color indices (L*, a*, b*), protein content, and muscle fiber diameter, did not differ significantly between groups (*p* > 0.05), indicating relatively consistent overall meat quality characteristics aside from lipid content.

Fatty acid profiling revealed several key differences (Table 1). The H group had significantly higher proportions of oleic acid (C18:1n9, *p* = 0.0016), arachidonic acid (C20:4n6, *p* = 0.0023), and total UFAs (57.31% vs. 52.08%, *p* = 0.0001), including both monounsaturated (MUFAs, *p* = 0.0064) and polyunsaturated fatty acids (PUFAs, though not statistically significant, *p* = 0.1094). Conversely, saturated fatty acids (SFAs) were significantly lower in the H group (42.43% vs. 47.94%, *p* = 0.0001), particularly palmitic acid (C16:0) and stearic acid (C18:0).

These findings indicate that higher UFA content in lamb muscle is associated with distinct alterations in fatty acid composition—particularly enrichment of oleic acid and depletion of SFAs—without compromising most meat quality parameters. The reduced fat content and increased moisture in the H group may also reflect a shift in lipid deposition patterns.

### 3.2. Transcriptomic Analysis Between High- and Low-UFA Groups

To investigate the transcriptional differences in muscle tissue between high- and low-unsaturated fatty acid (UFA) groups, we performed differential gene expression and functional enrichment analyses. Hierarchical clustering revealed a clear separation in expression profiles between the two groups (Figure 2A), which was further supported by principal component analysis (PCA), where the first principal component (PC1) explained 49% of the total variance (Figure 2B).

A total of 111 differentially expressed genes (DEGs) were identified based on a threshold of fold change > 1.5 and p < 0.05. Compared to the low-UFA (L) group, 64 genes were significantly upregulated, and 47 genes were downregulated in the high-UFA (H) group (Figure 2C). Notably, genes such as FADD, MYH7, TMEM74, CSNK2A2, and CTSB were markedly upregulated in the high-UFA group, suggesting potential links to fatty acid composition.

Gene Ontology (GO) enrichment analysis of DEGs (Figure 2D) revealed distinct functional patterns. Upregulated genes were enriched in biological processes (BPs) associated with muscle development and contraction, including striated muscle tissue development, muscle tissue development, and cardiac muscle development. In contrast, downregulated genes were associated with metabolic processes such as mitochondrial respiratory chain, ATP synthesis coupled electron transport, and fatty acid metabolism. Regarding molecular functions (MFs), upregulated genes were involved in actin binding, myosin binding, and calcium-dependent protein binding, while downregulated genes were enriched in NADH dehydrogenase activity and mitochondrial proton transport. Cellular component (CC) analysis showed that upregulated genes were primarily localized to myofibrils, contractile fibers, and Z discs, whereas downregulated genes were associated with mitochondrial inner membrane protein complexes.

KEGG pathway enrichment (Figure 2E) further indicated that upregulated genes were significantly involved in pathways such as cardiac muscle contraction, calcium signaling, and PI3K-Akt signaling. Conversely, downregulated genes were enriched in fatty acid degradation, peroxisome, and oxidative phosphorylation pathways. These findings suggest that the high-UFA group exhibits enhanced muscle contraction and signaling activities, whereas the low-UFA group features more active fatty acid metabolism and mitochondrial functions.

Protein–protein interaction (PPI) network analysis (Figure 2F) revealed that upregulated genes formed a muscle structure and contraction-related module centered on MYH7, TNNT1, and TNNI2, while downregulated genes clustered around COX1, ND1, and UQCRC1, key components of the mitochondrial respiratory chain complex.

Collectively, the transcriptomic data suggest that muscles in the high-UFA group exhibit a more oxidative fiber phenotype, with enhanced structural remodeling and signal transduction, whereas those in the low-UFA group show more pronounced fatty acid and energy metabolism. These molecular mechanisms may underline the observed differences in intramuscular UFA content.

### 3.3. Metabolomic Differences in Muscle Tissues Between High- and Low-UFA Groups

To investigate the impact of varying levels of unsaturated fatty acids (UFAs) on muscle metabolite profiles, we conducted untargeted metabolomic profiling of muscle tissues from high-UFA (H) and low-UFA (L) groups. The results revealed significant differences in metabolite composition between the two groups.

As shown in Figure 3A, hierarchical clustering of differential metabolites demonstrated clear separation between H and L groups, with good within-group consistency. Several metabolites, including N-acetylneuraminic acid, nicotinamide, and 18-hydroxy-5Z,8Z,11Z,14Z-eicosatetraenoic acid, were significantly differentially expressed between the groups.

Principal component analysis (PCA) further confirmed the metabolic divergence (Figure 3B). The first two principal components (PC1 and PC2) accounted for 69.6% of the total variance (PC1: 53.3%, PC2: 16.3%). A clear separation of the H and L groups was observed, indicating substantial differences in muscle metabolite profiles.

Volcano plot analysis identified 25 significantly different metabolites (Figure 3C, Table 2), including compounds upregulated in the H group such as 2(1H)-pyridinone and N-acetylneuraminic acid, and metabolites downregulated in the L group such as NADH and 18-hydroxyeicosatetraenoic acid. Differential metabolites were screened based on fold change (FC > 1.5) and statistical significance (p < 0.05).

These differential metabolites span amino acids, fatty acids and their derivatives, and small molecule regulators, suggesting a metabolic reprogramming of lipid and amino acid metabolism in the H group. For example, glutamyllysine, Lys-Trp-Arg, and D-aspartic acid were significantly upregulated in the H group, indicative of enhanced amino acid metabolism. In contrast, metabolites such as 6-phospho-D-gluconate and 2′-deoxycytidine were enriched in the L group, potentially reflecting altered carbohydrate and nucleotide metabolism. Notably, lipid-derived signaling molecules such as Resolvin D2 methyl ester and Prostaglandin E2 methyl ester were differentially expressed in the H group, suggesting roles in modulating lipid accumulation and immune microenvironment in muscle tissue.

Regarding fatty acid-related metabolites, palmitic acid, pentadecanoic acid, and 3-hydroxytetradecanoic acid were elevated in the H group, indicating increased fatty acid synthesis and associated enzymatic activity. In contrast, prostaglandins D1 and F1α were enriched in the L group, suggesting differential activation of the cyclooxygenase/prostaglandin pathway in lipid metabolism.

Pathway enrichment analysis revealed that the differential metabolites were predominantly involved in aspartate metabolism, the malate–aspartate shuttle, glycerolipid metabolism, and the urea cycle (Figure 3D). Among these, aspartate metabolism and the malate–aspartate shuttle were most significantly enriched, implying potential roles in regulating intramuscular UFA content.

### 3.4. Validation by Quantitative PCR and Western Blotting

To verify the reliability of the transcriptomic screening results, five differentially expressed candidate genes were randomly selected for quantitative real-time PCR (qRT-PCR) analysis in muscle tissues from the high-UFA (n = 6) and low-UFA (n = 6) groups. The qRT-PCR results showed expression trends consistent with those observed in the RNA-seq data (Figure 4A), confirming the robustness and reliability of the transcriptomic analysis.

To further validate protein-level expression changes of key genes, we performed Western blotting for two genes closely associated with fatty acid metabolism. As shown in Figure 4B, the protein levels of these targets were significantly higher in the high-UFA group compared to the low-UFA group (n = 3), in agreement with the transcriptomic trends. These results further support the potential regulatory roles of these genes in intramuscular unsaturated fatty acid deposition.

### 3.5. Comparison of Muscle Fiber Composition Between Groups

Among the genes identified by transcriptome analysis, MYH7 attracted particular attention, as it is a well-known marker gene involved in slow-twitch muscle fiber formation. To determine whether muscle fiber type composition differed between the high- and low-UFA groups, we performed immunofluorescence staining on longissimus dorsi muscle sections to quantify the proportions of slow and fast fibers (Figure 5). The results showed that the proportion of slow-twitch fibers was significantly higher in the high-UFA group compared to the low-UFA group (*p* = 0.0101), consistent with the transcriptomic findings. This suggests that muscles with a higher content of slow fibers tend to have a higher concentration of unsaturated fatty acids.

### 3.6. GSEA Analysis of the MYH7 Gene

To further explore the potential mechanisms by which MYH7 influences the intramuscular unsaturated fatty acid (UFA) content, a gene set enrichment analysis (GSEA) was performed using MYH7 expression levels as the phenotype. The results revealed several significantly enriched pathways associated with differential expression of MYH7 (Figure 6).

Notably, pathways related to energy and substrate metabolism, including the citrate cycle (TCA cycle), starch and sucrose metabolism, cysteine and methionine metabolism, and ABC transporters, were significantly enriched in the MYH7 low-expression group. This suggests that downregulation of MYH7 may enhance fundamental metabolic processes in skeletal muscle, potentially contributing to lipid metabolism and UFA deposition.

Conversely, pathways involved in immune regulation, such as complement and coagulation cascades, antigen processing and presentation, and the cytosolic DNA-sensing pathway, were significantly enriched in the MYH7 high-expression group, indicating a possible link between MYH7 expression and immune response. Additionally, the phenylalanine metabolism pathway was upregulated in the MYH7 high-expression group, reflecting shifts in amino acid metabolism.

Taken together, these findings suggest that MYH7 may influence UFA content in skeletal muscle by modulating metabolic and immune-related pathways. The crosstalk between these biological processes could represent an important mechanism underlying intramuscular fatty acid composition.

## 4. Discussion

Unsaturated fatty acid (UFA) content is a key determinant of meat nutritional value, flavor attributes, and its impact on consumer health [34]. In this study, we conducted a comprehensive analysis of transcriptomic and metabolomic data from muscle tissues with high versus low UFA levels, which revealed candidate genes, signaling pathways, and metabolic features associated with UFA deposition. These findings provide new molecular insights into lipid metabolism regulation in ruminant skeletal muscle.

### 4.1. Potential Regulatory Networks Revealed by Differentially Expressed Genes

Differential expression analysis showed significant upregulation of genes involved in energy metabolism, mitochondrial function, and lipid homeostasis in the high-UFA group, suggesting their possible role in intramuscular lipid deposition. Among them, FADD (Fas-associated death domain protein) displayed the highest statistical significance. Traditionally recognized as a key adaptor in apoptosis, recent studies have highlighted its “non-canonical” roles in metabolic regulation. FADD has been shown to modulate adipocyte metabolic activity via the NF-κB and mTOR signaling pathways, thereby influencing lipid droplet formation, lipid storage, and tissue homeostasis [35,36]. Consistently, FADD-deficient mice exhibit reduced respiratory quotient, diminished adipose mass, and decreased expression of lipogenic genes [37], suggesting its potential regulatory role in muscle lipid metabolism.

Other DEGs such as MCUR1, DEGS2, APOO, and ADIG may also participate in lipid metabolic regulation. MCUR1 regulates mitochondrial Ca^2+^ uptake, which has been reported to affect fatty acid oxidation and energy balance [38]. DEGS2, a key enzyme in sphingolipid synthesis, contributes to membrane stability and lipid signaling [39]. APOO is implicated in mitochondrial lipid transport [40], whereas ADIG (adipogenin) has been reported as an adipocyte-enriched transmembrane protein whose expression is strongly induced during adipogenesis; ADIG knockdown impairs adipocyte differentiation in vitro, and Adig knockout mice exhibit reduced fat mass, altered lipid droplet formation and impaired leptin secretion, indicating a functional role in adipogenesis and lipid droplet biology [41]. Thus, their upregulation in the high-UFA group may contribute to enhanced lipid storage capacity in muscle tissue.

Bidirectional GO enrichment analysis showed that upregulated genes were mainly enriched in processes related to muscle development, myofibril assembly, and actin/myosin binding, consistent with previous reports linking structural muscle genes to lipid deposition [42]. Conversely, downregulated genes were enriched in fatty acid metabolism, mitochondrial respiratory chain function, NADH dehydrogenase activity, and ATP synthesis, suggesting a shift toward reduced oxidative metabolism. Similar metabolic remodeling favoring lipid accumulation over catabolism has been reported in livestock muscle with higher intramuscular fat content [43]. Together, these functional enrichments suggest that muscle tissues in the high-UFA group favor lipid storage over oxidative metabolism, creating a metabolic environment conducive to UFA accumulation and flavor compound formation.

### 4.2. MYH7 as a Key Link Between Slow-Twitch Fibers and Lipid Deposition

MYH7 was significantly upregulated in the high-UFA group, with a fold change even higher than FADD (3.76 vs. 2.46), drawing particular attention. MYH7 encodes β-myosin heavy chain, a molecular marker of slow-twitch muscle fibers, whose elevated expression is typically associated with enhanced oxidative metabolism [21]. Slow-twitch fibers are known to be rich in mitochondria and possess high fatty acid oxidation capacity [44]. Previous studies in cattle and pigs have demonstrated that muscle with higher slow fiber content tends to accumulate more intramuscular lipids, improving meat quality traits [45]. Increased MYH7 expression reflects a higher proportion of slow fibers, which are enriched in lipid droplets and lipid-metabolizing enzymes [46], favoring intramuscular UFA storage. Notably, both FADD and MYH7 are closely linked to mitochondrial function, suggesting they may co-regulate lipid droplet formation and the surrounding metabolic milieu during lipid accumulation.

To further explore potential mechanisms by which MYH7 influences UFA deposition, we conducted GSEA to characterize the broader metabolic context associated with MYH7 expression. Genes negatively correlated with MYH7 were significantly enriched in the TCA cycle, carbohydrate metabolism, and amino acid metabolism, pathways known to provide energy and precursors for lipid synthesis [47]. Conversely, genes positively correlated with MYH7 were enriched in immune-related pathways such as complement and coagulation cascades and antigen processing. Similar immune–metabolic crosstalk has been reported in skeletal muscle, where immune pathways influence lipid turnover and deposition [48,49]. This dual metabolic–immune regulation offers a novel perspective on how MYH7 may influence intramuscular fatty acid composition.

### 4.3. Metabolomic Characteristics Reveal the Biochemical Basis of Lipid Deposition

Metabolomic analysis revealed significant upregulation of various lipid and amino acid metabolites in the high-UFA group. Elevated levels of N-acetylneuraminic acid and nicotinamide are consistent with enhanced NAD^+^-dependent energy metabolism and mitochondrial activity [50], which may promote lipid synthesis or inhibit lipolysis. Upregulated lipid metabolites, including stearic acid, pentadecanoic acid, and 3-hydroxytetradecanoic acid, indicate increased activity of fatty acid biosynthesis pathways [51].

Pathway enrichment analysis further indicated involvement of aspartate metabolism, the malate–aspartate shuttle, triglyceride metabolism, and the urea cycle. These pathways have been reported to provide carbon skeletons and reducing equivalents for fatty acid synthesis in ruminant muscle [52].

Moreover, the upregulation of lipid signaling molecules such as resolvin D2 methyl ester and prostaglandin E2 methyl ester in the high-UFA group may be involved in modulating lipid accumulation and local immune status. These findings align with the immune pathway enrichment observed in MYH7-high samples, suggesting that muscle immune environments may influence lipid homeostasis via metabolic signaling.

### 4.4. Limitations and Future Perspectives

This study, based on RNA-seq and untargeted metabolomics, systematically identified candidate genes and pathways associated with differential UFA content and proposed a potential regulatory mechanism involving the interplay between muscle structure, metabolism, and immunity. However, several limitations remain: the findings are correlative and require functional validation (e.g., gene knockout/overexpression, lipid droplet staining, or metabolic flux analysis); the use of sheep as the model species raises questions about the translatability of the results to other livestock such as pigs, cattle, or poultry; and whether shifts in muscle fiber type are causal to UFA deposition or merely associative needs to be addressed in future longitudinal or interventional studies.

## 5. Conclusions

By integrating transcriptomic, metabolomic, and muscle phenotypic data, this study systematically elucidates the molecular basis underlying differences in unsaturated fatty acid (UFA) content in the longissimus dorsi muscle of sheep. Individuals with higher UFA content exhibited distinct transcriptional signatures associated with lipid metabolism, mitochondrial function, and immune responses. Metabolomic profiling further confirmed the accumulation of metabolites related to both fatty acid synthesis and degradation. Notably, MYH7, a hallmark gene of slow-twitch muscle fibers, was upregulated in the high-UFA group, suggesting that muscle fiber-type transition may contribute to UFA deposition. As a potential regulatory nexus linking muscle fiber composition and fatty acid metabolism, MYH7 may play a key role in promoting UFA accumulation in muscle. These findings offer molecular insights and potential targets for improving the fatty acid profile and nutritional quality of lamb meat.

## Figures and Tables

**Figure 1 animals-15-02617-f001:**
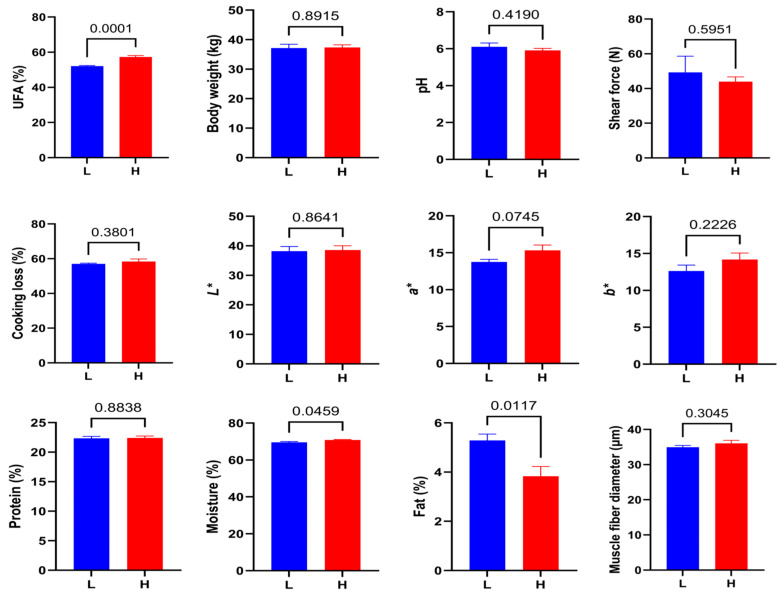
Comparison of unsaturated fatty acid (UFA) content, body weight, and meat quality traits between lambs with high UFA content (H group) and low UFA content (L group). Meat quality traits include ultimate pH (pH value of muscle measured 24 h postmortem), tenderness (shear force, N), cooking loss (%), meat color parameters (L*, lightness; a*, redness; b*, yellowness), protein content (%), moisture content (%), intramuscular fat content (%), and muscle fiber diameter (µm). Data are presented as mean ± SEM. *p* < 0.05 was considered statistically significant.

**Figure 2 animals-15-02617-f002:**
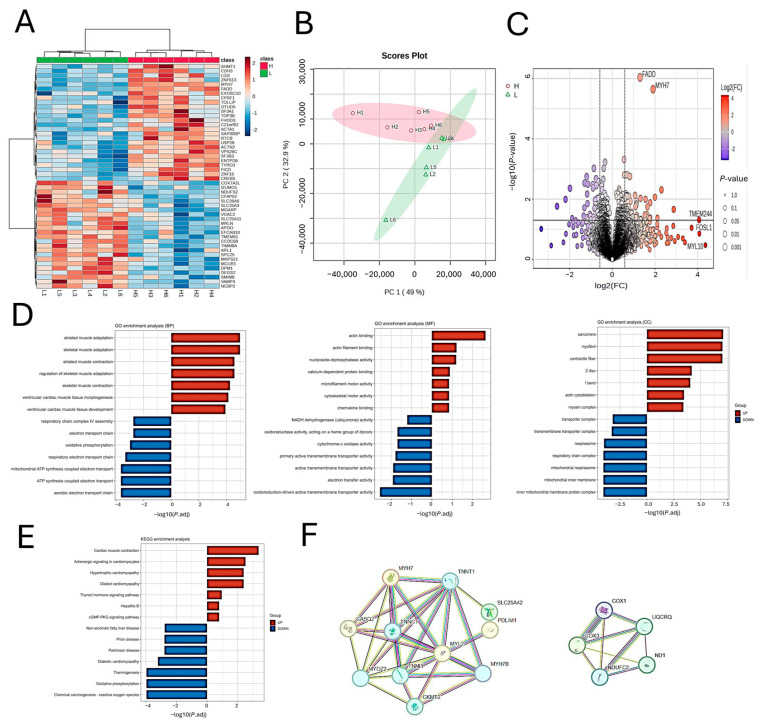
Transcriptomic analysis of longissimus dorsi muscle tissues from lambs with high (H) and low (L) UFA content. (**A**) Heatmap showing hierarchical clustering of differentially expressed genes (DEGs). Each column represents one sample, and each row represents one gene. Red indicates high expression and blue indicates low expression. (**B**) Principal component analysis (PCA) plot showing separation between H and L groups based on transcriptomic profiles (PC1 and PC2 represent the first and second principal components). (**C**) Volcano plot of DEGs. The x-axis shows log_2_(fold change, FC), and the y-axis shows −log_10_(p value). Red dots represent significantly upregulated genes, and blue dots represent significantly downregulated genes (FC > 1.5, *p* < 0.05). (**D**) Gene Ontology (GO) enrichment analysis of DEGs. The top enriched terms are shown in the biological process (BP), molecular function (MF), and cellular component (CC) categories. (**E**) Kyoto Encyclopedia of Genes and Genomes (KEGG) pathway enrichment analysis of DEGs. (**F**) Protein–protein interaction (PPI) network of DEGs. Each node represents a gene, and the edges indicate predicted interactions.

**Figure 3 animals-15-02617-f003:**
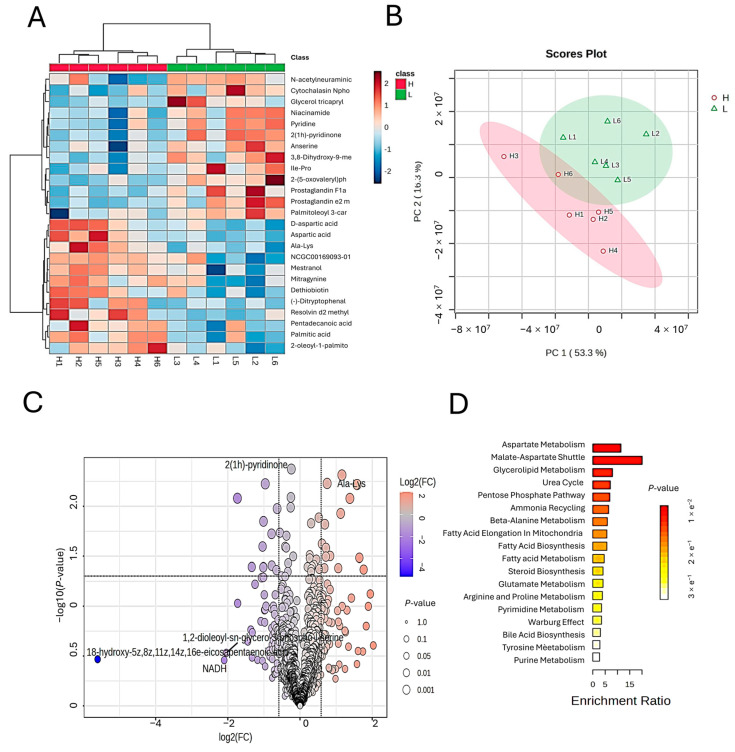
Metabolomic profiling of muscle tissues from lambs with high (H) and low (L) UFA content. (**A**) Heatmap of differentially expressed metabolites (DEMs). Each column represents a sample, and each row represents a metabolite. Red indicates high abundance, and blue indicates low abundance. (**B**) PCA plot showing clustering of samples (H vs. L) based on metabolic profiles (PC1 and PC2 represent the first and second principal components). (**C**) Volcano plot of DEMs. The x-axis shows log_2_(FC), and the y-axis shows −log10(p value). Red dots indicate significantly upregulated metabolites, and blue dots indicate significantly downregulated metabolites (FC > 1, *p* < 0.05). (**D**) KEGG pathway enrichment analysis of DEMs.

**Figure 4 animals-15-02617-f004:**
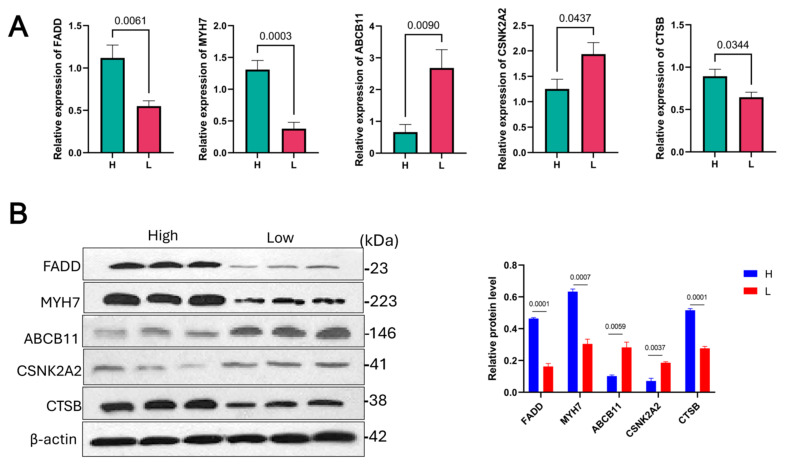
Validation of differentially expressed genes between high (H) and low (L) UFA content groups using qRT-PCR and Western blot analysis. (**A**) Quantitative real-time PCR (qRT-PCR) analysis of five randomly selected DEGs (n = 6 per group). Relative mRNA levels were normalized to β-actin expression. (**B**) Western blot analysis of protein expression levels (n = 3 per group). Band intensities were quantified and expressed as mean ± SEM. *p* < 0.05 was considered statistically significant.

**Figure 5 animals-15-02617-f005:**
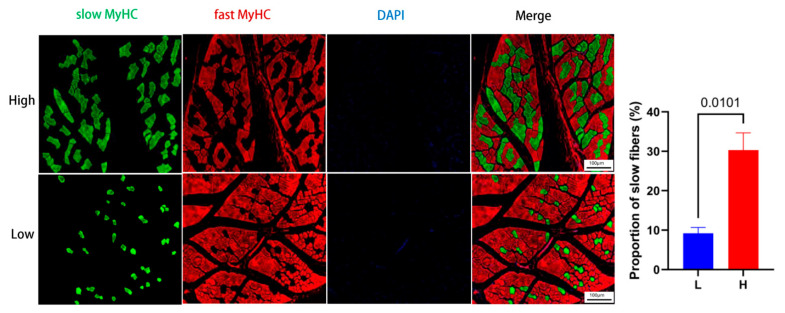
Immunofluorescence staining of muscle fiber types in longissimus dorsi from high (H) and low (L) UFA content groups. Muscle sections were stained with antibodies against slow myosin heavy chain (slow MyHC, green) and fast myosin heavy chain (fast MyHC, red). Cell nuclei were counterstained with DAPI (blue). Representative images are shown (scale bar = 100 μm). Quantitative results are expressed as mean ± SEM (n = 3 per group).

**Figure 6 animals-15-02617-f006:**
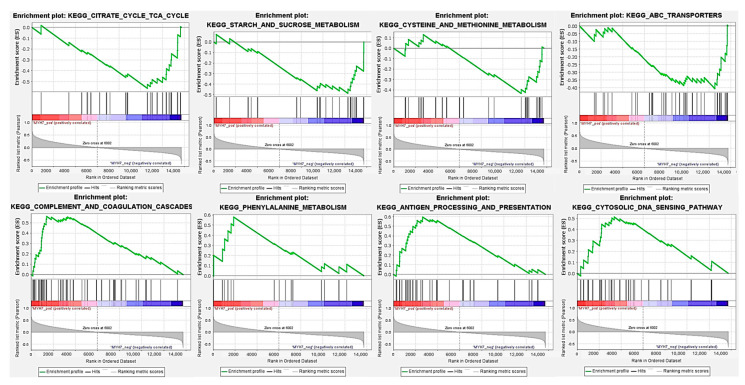
Gene Set Enrichment Analysis (GSEA) of MYH7 in skeletal muscle. Enrichment plots display the distribution of genes ranked by expression level, with curves representing the enrichment score (ES) for significantly associated biological processes and signaling pathways. *p* < 0.05 was considered statistically significant.

**Table 1 animals-15-02617-t001:** Comparison of fatty acid composition (% of total fatty acids) between different groups.

Fatty Acid	L	H	*p*-Value	Cohen’s d
Oleic acid (C18:1n9)	44.17 ± 0.57	47.73 ± 0.50	0.0016	2.17
Arachidonic acid (C20:4n6)	1.18 ± 0.05	1.93 ± 0.19	0.0023	1.73
Palmitic acid (C16:0)	25.27 ± 0.47	23.05 ± 0.54	0.0128	1.4
Stearic acid (C18:0)	19.55 ± 0.59	17.27 ± 0.58	0.0572	1.29
Myristic acid (C14:0)	2.08 ± 0.09	1.63 ± 0.13	0.0522	1.33
Heptadecanoic acid (C17:0)	0.89 ± 0.07	0.69 ± 0.04	0.0797	1.17
Linoleic acid (C18:2n6)	4.78 ± 0.32	5.56 ± 0.52	0.1801	0.63
Palmitoleic acid (C16:1)	1.61 ± 0.10	1.68 ± 0.11	0.738	0.21
Pentadecanoic acid (C15:0)	0.19 ± 0.01	0.11 ± 0.03	0.0601	1.22
cis-10-Heptadecenoic acid (C17:1)	0.61 ± 0.03	0.55 ± 0.02	0.1732	0.83
cis-10-Pentadecenoic acid (C15:1)	0.07 ± 0.03	0.03 ± 0.02	0.4143	0.57
Saturated Fatty Acids (SFAs)	47.94 ± 0.35	42.43 ± 0.80	0.0001	3.44
Unsaturated Fatty Acids (UFAs)	52.08 ± 0.36	57.31 ± 0.79	0.0001	3.48
Monounsaturated Fatty Acids (MUFAs)	46.46 ± 0.74	49.99± 0.71	0.0064	1.98
Polyunsaturated Fatty Acids (PUFAs)	5.62 ± 0.40	7.31 ± 0.87	0.1094	1.01

H: high UFA content group, L: low UFA content group; data are presented as mean ± SEM. *p* < 0.05 was considered statistically significant.

**Table 2 animals-15-02617-t002:** Differential metabolites between high- and low-UFA muscle groups.

	FC	log_2_(FC)	*p*-Value
Ala-Lys	2.235	1.1603	0.0048582
Palmitic acid	1.6827	0.75082	0.0058927
Prostaglandin e2 methyl ester	0.51287	−0.96334	0.0059498
Aspartic acid	2.9882	1.5793	0.0060464
2-(5-oxovaleryl)phosphatidylcholine	0.30046	−1.7348	0.008306
Resolvin d2 methyl ester	2.6036	1.3805	0.0083364
3,8-Dihydroxy-9-methoxypterocarpan	0.64406	−0.63474	0.010545
D-aspartic acid	2.1818	1.1255	0.011801
Prostaglandin F1a	0.4956	−1.0127	0.014185
NCGC00169093-01	1.6099	0.68698	0.015147
Glycerol tricaprylate	0.57869	−0.78914	0.018894
Pentadecanoic acid	1.524	0.60791	0.023797
Cytochalasin Npho	0.48678	−1.0387	0.025344
Dethiobiotin	1.6431	0.7164	0.026165
(-)-Ditryptophenaline	1.7832	0.83444	0.031486
Isoanhydroicaritin	1.6412	0.71477	0.031982
Lys-Trp-Arg	3.1238	1.6433	0.032825
2′-deoxycytidine	0.58106	−0.78323	0.039292
N6-2-(4-aminophenyl)ethyladenosine	0.3957	−1.3375	0.040449
6-Phospho-D-gluconate	0.49474	−1.0153	0.040617
Prostaglandin d1	0.4935	−1.0189	0.040847
Ciprofloxacin piperazinyl-n4-sulfate	2.0831	1.0587	0.0416
3-Hydroxytetradecanoic acid	1.5959	0.67441	0.041679
Glutamyllysine	3.391	1.7617	0.042961
Idebenone	0.63487	−0.65546	0.043434

Differential metabolites were screened based on fold change (FC > 1.5) and statistical significance (*p* < 0.05).

## Data Availability

Primer sequences used for RT-qPCR experiments are provided in Appendix A. RNA-seq data are included in NCBI Sequence Read Archive (SRA) PRJNA1139683. Other data will be shared on reasonable request to the corresponding author.

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
