# Peer review of "Molecular Insights into Intramuscular Unsaturated Fatty Acid Deposition in Lambs Through Multi-Omics Profiling"

_animals, 2025, doi:10.3390/ani15172617_

Round 1
Reviewer 1 Report
Comments and Suggestions for Authors
The topic addressed in this paper is interesting and relevant for understanding the molecular insights into intramuscular unsaturated fatty acid deposition in lambs, however, the study requires several methodological and clarity improvements in the presentation of results before it can be considered for publication.
Introduction:
! It would be useful to add 2-3 paragraphs in the introduction to express whether such studies have been done on lambs based on Multi-Omics Profiling and what the results were.
Materials and Methods
! For each analysis program/software used, please provide the name of the software, its version number, and the developer or provider. Just example: SPSS Statistics 25.0 (Armonk, NY, USA)
! Figures and tables are cited in the text (Figure 1), (Figure 2), (Table 1), (Table 2) etc...not (Fig. 1) or (Tab. 1). Proofread throughout the manuscript!
! Ë®protocols followed those described by Han et al. [22].Ë®…. it's not enough to cite a source.. go into detail
! It would be more useful if the data regarding the primers used in the PCR analysis, as well as the PCR amplification temperatures and times, appeared in the manuscript in the Material and Method section.
Results
! Figures 2, 3 and 6 are unclear! Please review this!
Discussion
!The Discussion section should be improved by incorporating data from recent and relevant studies in the specialized literature, in order to better support and contextualize the findings.
References
!The journal's template has not been followed in the formatting of the references. Please correct this aspect so that all citations comply with the style required by the journal.
!36 is not a bibliographical source...be careful!
Reviewer 2 Report
Comments and Suggestions for Authors
In this study, the authors explained the molecular insights about the deposition of intramuscular unsaturated fatty acids through multi-omics in lambs. The article as a whole is well written, and all data related to this study are also presented in either supplementary data form or through the NCBI repository. But still, there are some issues that need to be addressed before moving forward with publication.
- Lines 18, 29, 32, 77, 78, 79, 80, 84, and so on across the manuscript are all gene symbols that need to be italicized, considering the expression level of these genes.
- Lines 45-47, 53-54, and 67-69, restructure and reword for better understanding.
- Lines 77-106 are part of the results and discussion, so should not be included in the introduction.
- Lines 139, please highlight which RNA-Seq, like mRNA seq, or others.
- Include the version and citation for tools used, such as for HISAT2 and DESeq2 for lines 142 and 144.
- Also provide the reference genome used for enrichment analysis in ClusterProfiler at line 146, along with version and citation.
- Line 159 shows MetaboAnalyst6.0, and it should be like MetaboAnalyst v6.0.
- Line 178 for supplementary table S2 lacks the catalog ID of the antibodies.
- Line 183 showed a different tool for enrichment analysis. Why didn't we use the same as before, like ClusterProfiler.
- Legend details for Figure 1, Table 1, and so on across the manuscript are insufficient to explain the acronyms and x-to-y-axis details of the graph. These figures and tables should be self-explanatory.
- Line 380 mentioned integrated analysis, and as per my understanding, the integration approach involved the pattern development of multi-omics datasets.
- Lines 397-409, and 421-448, lack proper citations to support the discussion, but what I see is that the authors again discussed the results in the discussion section instead of the discussion section, with support from previous literature cited accurately.
Round 2
Reviewer 1 Report
Comments and Suggestions for Authors I agree with all the improvements made to the manuscript after the first revision, but for publication, I recommend that authors carefully revise the reference section according to the Animals guidelines, following the specific format for articles, book chapters, and books, as illustrated in the examples provided: 1. Author 1, A.B.; Author 2, C.D. Article title. Abbreviated name of the journal Year, Volume, page range. 2. Author 1, A.; Author 2, B. Chapter title. In Book title, 2nd edition; Editor 1, A., Editor 2, B., Eds.; Publisher: Publisher location, Country, 2007; Volume 3, pp. 154–196. 3. Author 1, A.; Author 2, B. Book title, 3rd edition; Publisher: LocaÈ›ia publishing house, Èšara, 2008; pp. 154–196. and so on...Author Response
Please see the attachment.

Reviewer 2 Report
Comments and Suggestions for Authors
The edited version of the manuscript is in better shape, and as a reviewer, I really appreciate the efforts of the authors in this regard. But there are some comments that still need the attention of the author(s) before moving forward with acceptance of this updated draft.
- Lines 45-47, 53-54, and 67-69, restructure and reword for better understanding.
These sentences are still needed to be restructured and reworded now at lines 47-49, 53-55, and 68-70.
- Lines 77-106 are part of the results and discussion, so should not be included in the introduction.
Although, authors tried to improve this section and I really appreciate the efforts of the authors. But this section of introduction lacks proper citation to facts described i.e., lines 77-79, 85-87.
- Legend details for Figure 1, Table 1, and so on across the manuscript are insufficient to explain the acronyms and x-to-y-axis details of the graph. These figures and tables should be self-explanatory.
Figures 4 and 5 still lack the description for acronyms like high (H) and Low (L).
